# Natural Disease Course in Usher Syndrome Patients Harboring *USH2A* Variant p.Cys870* in Exon 13, Amenable to Exon Skipping Therapy

**DOI:** 10.3390/genes14030652

**Published:** 2023-03-05

**Authors:** Katja Čadonič, Jana Sajovic, Marko Hawlina, Ana Fakin

**Affiliations:** 1Eye Hospital, University Medical Centre Ljubljana, 1000 Ljubljana, Slovenia; 2Faculty of Medicine, University of Ljubljana, 1000 Ljubljana, Slovenia

**Keywords:** Usher syndrome, *USH2A*, retinitis pigmentosa, fundus autofluorescence, exon skipping therapy

## Abstract

The aim of the study was to determine the rate of retinal degeneration in patients with c.2610C>A (p.Cys870*) in *USH2A* exon 13, amenable to exon skipping therapy. There were nine patients from seven families, three of whom were male (two were homozygous). Seven patients had follow-up data (median of 11 years). Analysis included best corrected visual acuity (BCVA, decimal Snellen), visual field (Goldmann perimetry target II/4), fundus autofluorescence (FAF), optical coherence tomography (OCT), and microperimetry (MP). The median age at the onset of nyctalopia was 20 years (range, 8–35 years of age). At the first exam, at a median age of 42 years, the median BCVA was 0.5 (0.2–1.0), and the median visual field diameter was 23° (5°–114°). Imaging showed a hyperautofluorescent ring delineating preserved foveal photoreceptors in 78% (7/9) of patients, while 22% (2/9) had a hyperautofluorescent patch or atrophy, reflecting advanced disease. Survival analysis predicted that 50% of patients reach legal blindness based on a visual field diameter < 20° at the age of 52 (95% CI, 45–59) and legal blindness based on a BCVA ≤ 0. 1 (20/200) at the age of 55 (95% CI, 46–66). Visual field constriction occurred at the median rate of radial 1.5 deg/year, and hyperautofluorescent ring constriction occurred at the median rate of 34 μm/year. A non-null second allele was found in two patients: p.Thr4315Pro and p.Arg303His; the patient with p.Arg303His had a milder disease. The rates of progression will be useful in the design and execution of clinical trials.

## 1. Introduction

Usher syndrome is an autosomal recessive genetic disease presenting with a combination of retinitis pigmentosa (RP) and varying degrees of hearing loss. Among the three major clinical subtypes, Usher syndrome type II is the most frequent, caused by pathogenic variants in *USH2A*, *GPR98,* and *DFNB31,* and is characterized by moderate to severe congenital hearing loss, the onset of RP in the second decade of life, and normal vestibular function [1,2].

Causative variations most frequently occur in the *USH2A* gene, which consists of 72 exons and encodes the structural protein Usherin, crucial for retinal photoreceptors and cochlear hair cells [3,4,5]. Impairment of Usherin results in early histopathological findings in the retina that include shortening of rod outer segments and cell death. These changes lead to problems with night vision, or nyctalopia. Later, cone cells are also damaged and slowly degenerate, causing central vision loss [3,6]. While hearing loss can be treated with hearing aids or cochlear implants, visual loss is currently untreatable and ends with legal blindness in the majority of patients [7,8]. Genetic therapy is under development for certain subgroups of patients, with the aim of slowing disease. We studied nine patients with a c.2610C>A, p.(Cys870*) variant in exon 13 of the *USH2A* gene who would be amenable to exon skipping therapy with QR-421a, currently in phases 2 and 3 of clinical trials (Clinicaltrials.gov: NCT05158296; NCT05158296). Treatment using this approach has shown promising results in zebrafish $ush2a^{rmc1}$ mutants, achieving a partially restored expression of Usherin protein in photoreceptors [9]. Knowledge of natural disease progression will be important for the execution of clinical trials in humans.

## 2. Results

The summary of genetic and phenotypic characteristics is shown in Table 1 and Table 2, respectively. The complete dataset of all measured clinical data is available in Appendix A.

### 2.1. Disease Onset

The median age when patients first noted difficulties with night vision was 20 years (range, 8–35; N = 8 with available data). The two patients with a non-null genotype (patients 3 and 4) had onsets at 21 and 23 years of age, respectively.

### 2.2. Visual Acuity and Visual Field

Deterioration of visual acuity and visual field with time is shown in Figure 1 and Figure 2, respectively. At the first exam, at the median age of 42 (range, 25–56), the median BCVA of the better-seeing eye was 0.3 logMAR (0.5 Snellen decimal). Among the six patients with follow-up (excluding patient 8, whose vision improved after cataract surgery), the median yearly loss of visual acuity was estimated at 0.03 logMAR, or 5.3% per year. The median yearly loss of visual field diameter was estimated at 1.5° (range, 0.1–5.6°) or 3.7% (range, 0.5–8.7%).

**Figure 1 genes-14-00652-f001:**
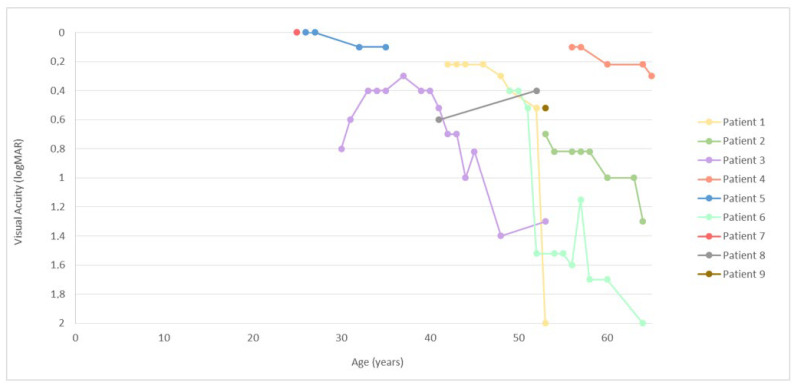
Best-corrected visual acuity of the right eye in relation to age. Note a relatively sharp decrease in visual acuity around the age of 50. Among patient with a non-null second allele, better acuity was observed in patient 4 (p.Arg303His), but not in patient 3 (p.Thr4315Pro). The increase in visual acuity of patient 8 was related to cataract surgery.

**Figure 2 genes-14-00652-f002:**
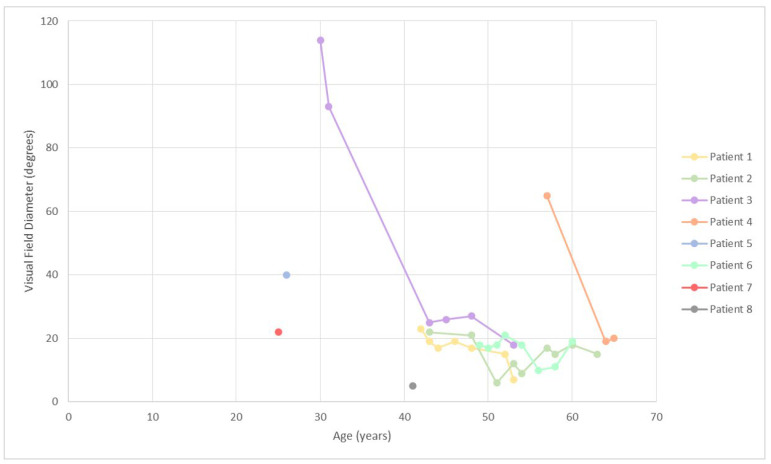
Visual field diameter (Isopter II/4) of the right eye in relation to age. Note the relatively larger visual fields in patients 3 and 4 harboring missense variants at the first exam; however, only patient 4 (p.Arg303His) retained the advantage with follow-up. There is an apparent improvement in the visual field of patient 2 after previous worsening. The patient had extensive cystoid macular edema CME around the age of 50 (Figure 3), which resolved with time and may have contributed to the variability of the visual field.

**Figure 3 genes-14-00652-f003:**
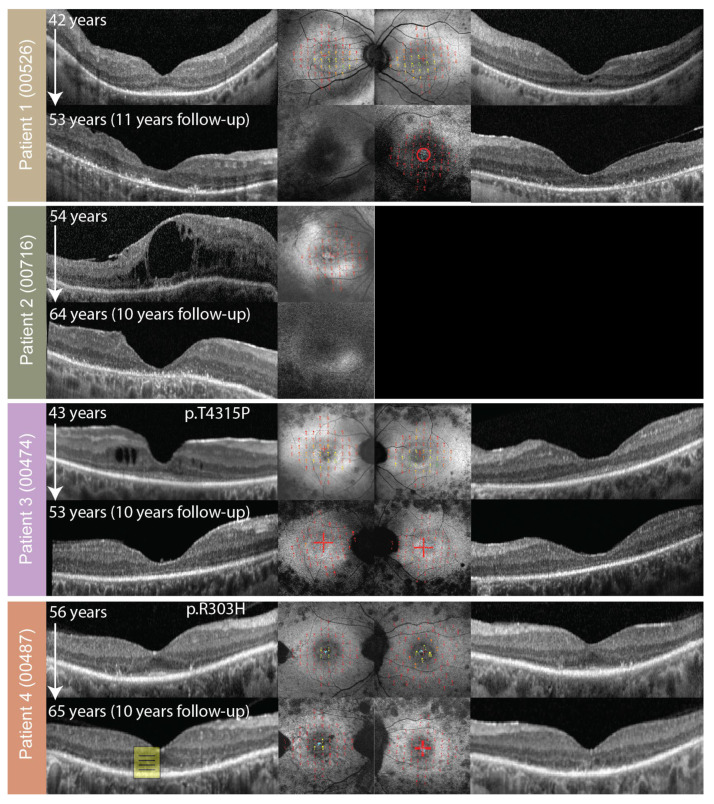
OCT (side columns) and FAF with overlaid microperimetry (the central two columns) of patients 1–4. The right eye is shown on the left and the left eye on the right. Patient ID (center ID) is stated on the left side. Follow-up images are shown if available. Patient 2 had undergone enucleation of the left eye for unrelated reasons. Microperimetry is on a color scale from good sensitivity (green) to poor retinal sensitivity (red). Fixation points are marked with blue dots. Note good retinal sensitivity inside the hyperautofluorescent ring in patients with this FAF pattern (patients 1, 3, and 4) and loss of retinal sensitivity with progression to the hyperautofluorescent patch on follow-up (patients 1 and 3).

Based on the data from the better eye, Kaplan Maier analysis predicted that 50% of patients would reach s visual field diameter < 20° at the age of 52 (95% CI, 45–59), BCVA ≤ 0.1 (≥1.0 logMAR) at the age of 55 (95% CI, 44–66), and BCVA ≤ 0.05 at the age of 58 (95% CI, 52–63) (Figure 4). Legal blindness was predicted for 50% patients by the age of 50 (range, 43–58) based on either visual acuity (≤0.1) or visual field (<20°).

**Figure 4 genes-14-00652-f004:**
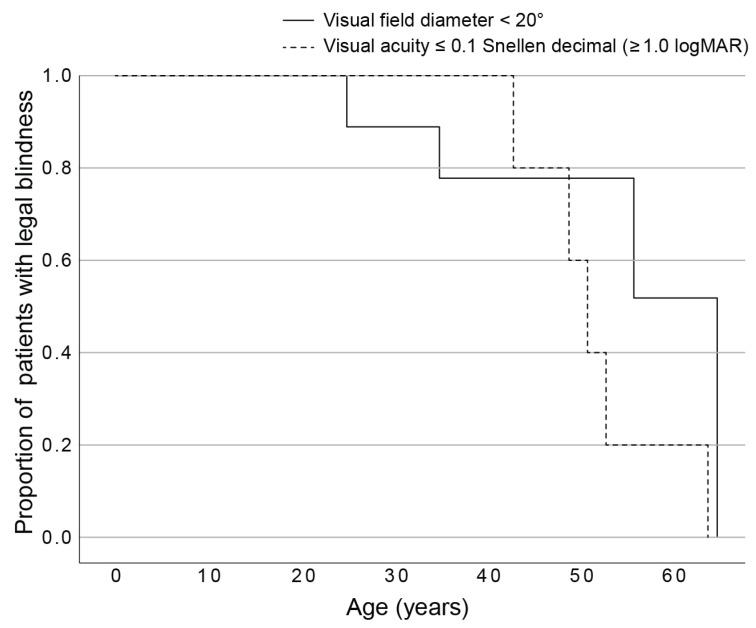
Kaplan Meier survival analysis plot showing the percentage of patients reaching legal blindness based on visual field (solid line) or visual acuity (dashed line).

### 2.3. Fundus Autofluorescence and Optical Coherence Tomography

All patients underwent FAF and OCT imaging, among whom seven were followed longitudinally (Figure 5). At the first imaging, at the median age of 43 (range, 25–56), 7/9 (78%) patients had bilateral hyperautofluorescent rings; one (patient 2, 54 years old) had a hyperautofluorescent patch on her only eye; and one (patient 6, 49 years old) had a combination of a patch and atrophy. Patient 4 exhibited a second hyperfluorescent ring in the periphery, which corresponded with preserved peripheral retina and is described in detail in a previous publication [10]. Among the patients with follow-up, five patients started with rings, which all showed constriction, and in two cases (patients 1 and 3), progressed to a hyperautofluorescent patch (at ages 49 and 53, respectively). Patient 2 had a patch at their first imaging at the age of 54 and progressed to atrophy on follow-up. The rate of ring diameter constriction in the five patients with rings who had follow-up (median 9 years, range 2–11 years) was calculated to be at the median of 34 μm per year (range, 10–99), or, considering the diameter at first imaging, 2.4% per year (range, 0.6–5.1%). Notable cystoid macular oedema was present in 2 out of 9 patients (22%), was present at the first exam, and resolved with follow-up in both patients.

**Figure 5 genes-14-00652-f005:**
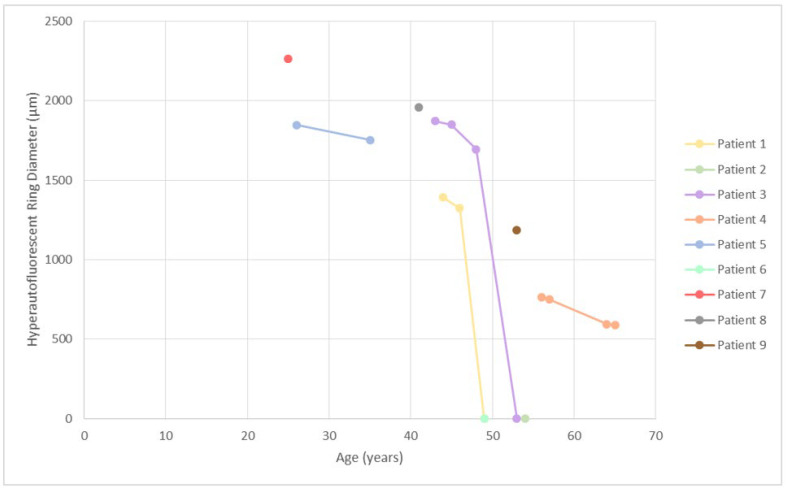
Hyperautofluorescent ring diameter in association with age. Patients 2 and 3 had hyperautofluorescent patches and/or atrophy; therefore their diameters are marked as zero. Note the trend of progression from ring to patch/atrophy around the age of 50, which corresponds to the sharp decrease in visual acuity (Figure 1 and Figure 4).

### 2.4. Microperimetry

Microperimetry results are shown in Figure 3 and Figure 6, overlaid over FAF. At the first exam, at the median age of 43 (range, 25–57 years of age; N = 8 with available data), the median MS (mean sensitivity) in the macula was 1.20 dB (range, 0.00–2.60 dB). The patients that were exhibiting hyperautofluorescent rings (patients 1, 3, 4, 5, 7, and 8), fixated with fovea and had preserved retinal sensitivity inside the hyperautofluorescent rings. Patients 2 and 6 (aged 54 and 56, respectively) had no detectable retinal sensitivity at the first microperimetry exam. The median yearly loss of MS (calculated for four patients with follow-up) was estimated at 0.16 dB (range, 0.06–0.24 dB) or 7.50% (range 6.88–8.89%). Loss of sensitivity was most notable in patients 1 and 3 who experienced a transition from rings to patches/atrophy on FAF (Figure 3).

**Figure 6 genes-14-00652-f006:**
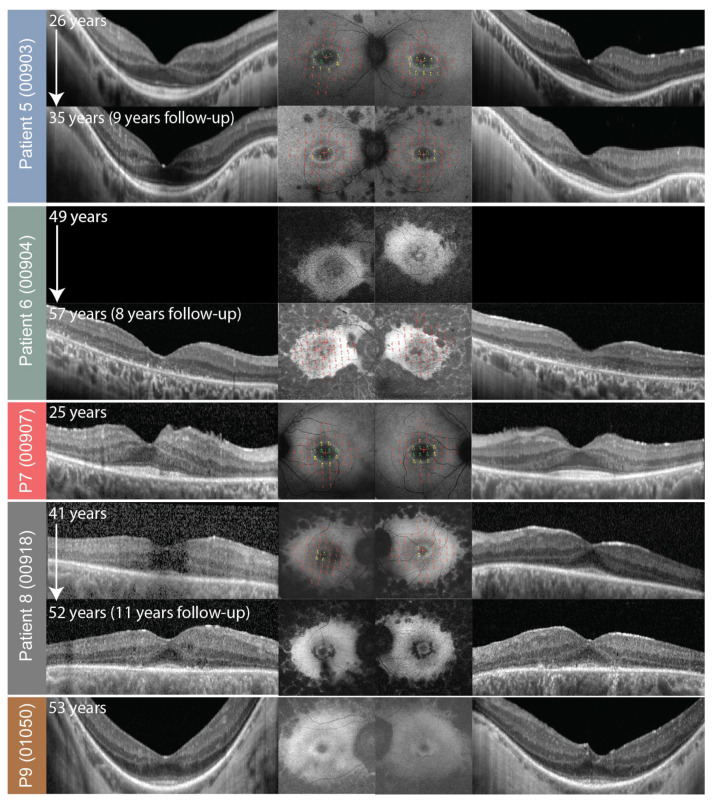
OCT (side columns) and FAF with overlaid microperimetry (central two columns) of patients 5–9. The right eye is shown on the left and the left eye on the right. Patient ID (center ID) is stated on the left side. Follow-up images are shown if available. Microperimetry images are enlarged in Figure 7 and Figure 8.

**Figure 7 genes-14-00652-f007:**
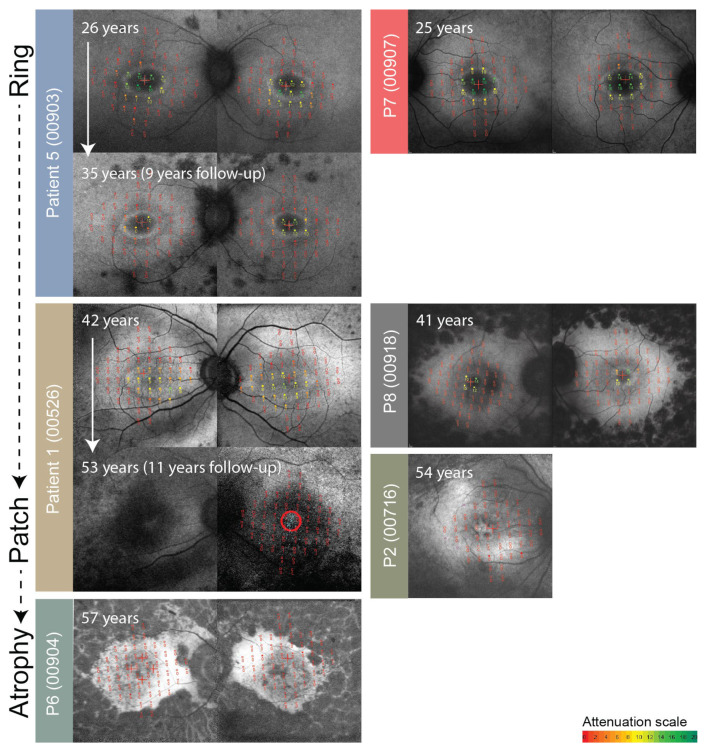
Enlarged FAF with overlaid microperimetry of patients who had p.(Cys870*) in *trans* with a stop allele. The right eye is shown on the left and the left eye on the right for each patient. Patient ID (center ID) is stated on the left side of each eye pair. Follow-up is shown when available. Microperimetry results are on a color scale from good sensitivity (green) to poor retinal sensitivity (red), with the attenuation scale in dB shown below. The patients are arranged from youngest to oldest in a vertical direction to demonstrate the progression of disease from the early stage (hyperautofluorescent ring) to the advanced stage (patch and atrophy). Note preserved retinal sensitivity inside the hyperautofluorescent ring and severe loss of retinal sensitivity in advanced stages.

**Figure 8 genes-14-00652-f008:**
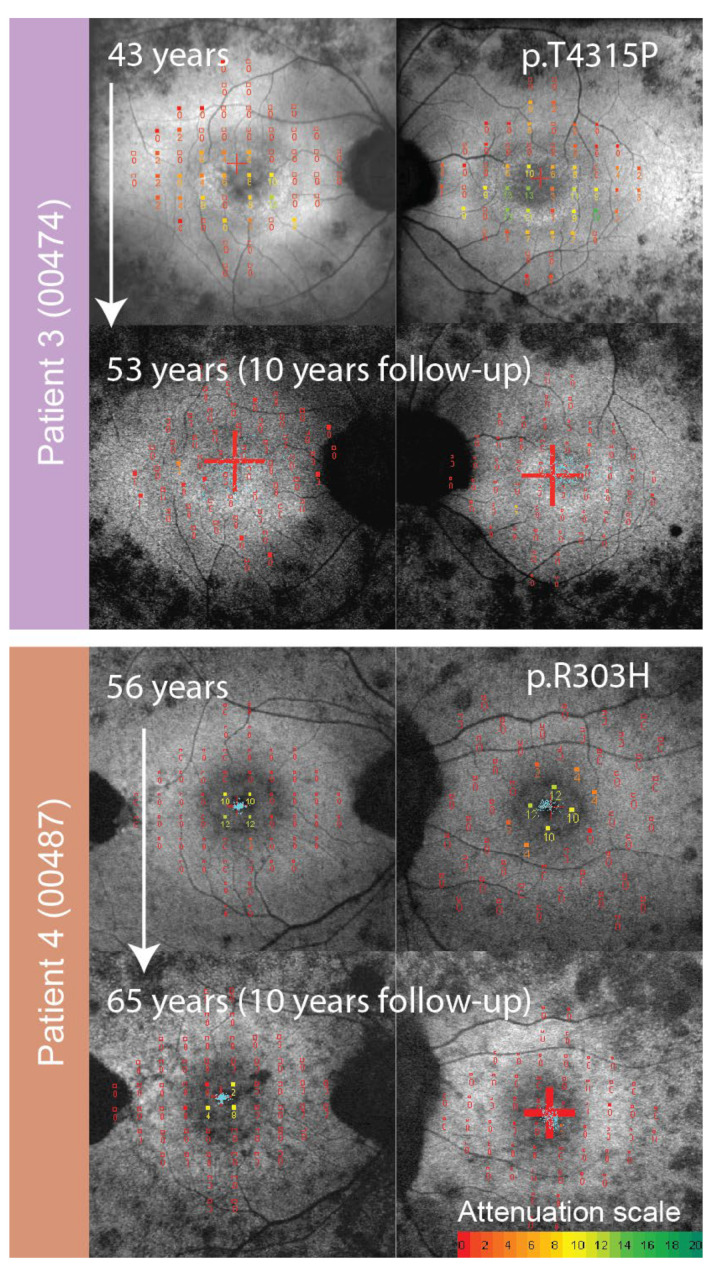
Enlarged FAF with overlaid microperimetry of patients who had p.(Cys870*) in *trans* with a missense allele. The right eye is shown on the left and the left eye on the right for each patient. Patient ID (center ID) is stated on the left side. Follow-up images are shown for both patients. Microperimetry results are on a color scale from good sensitivity (green) to poor retinal sensitivity (red), with the attenuation scale in dB shown below. Note the relatively well preserved central retinal function in Patient 4 harboring p.(Arg303His) in comparison to patients with two stop alleles of a similar age (Figure 7).

### 2.5. Electrophysiology

Five patients underwent full-field electrophysiology (ffERG) testing. The ffERG amplitudes in all cases were consistent with retinitis pigmentosa. They were reduced in the pattern of rod-cone dystrophy in two patients (P4 and P5; aged 52 and 26, respectively). and undetectable in three patients (P1, P6, and P7; aged 42, 50, and 25, respectively). The oldest patient with detectable ffERG was patient 4, at the age of 52, who harbored p.(Arg303His) on the second allele (Figure 9). Patients P1, P4, and P5 also underwent pattern ERG (PERG) and visual evoked potentials (VEP) at ages 42, 52, and 26, respectively. All had reduced PERG P50 amplitude and normal VEP amplitude and latency. Their visual acuity at the time of the recording was 0.5–1.0 Snellen decimal, and all three exhibited a hyperautofluorescent ring on the FAF at that time (or on the first occasion that imaging was performed, if performed subsequently).

## 3. Discussion

This study defined the natural course of retinal disease in *USH2A* patients harboring p.(Cys870*) in *USH2A* exon 13 over a long-term follow-up, which is of importance in the design of potential clinical trials involving exon 13 skipping.

### 3.1. Disease Course Associated with USH2A p.(Cys870*) with Implications for Clinical Trials

Disease in patients with p.Cys870* followed the previously defined structural benchmarks on fundus autofluorescence, starting with a hyperautofluorescent ring and progressing to patches and atrophy [11]. The visual function deteriorated accordingly, starting with constriction of the visual field and followed by a sharp decline in visual acuity when the ring converted to a patch and loss of foveal photoreceptors occurred. The majority of patients reached legal blindness due to the constriction of their visual field. Specifically, Kaplan Maier analysis predicted that 50% of patients reached a visual field diameter < 20° at the age of 52, BCVA ≤ 0.1 at 55 years of age, and BCVA ≤ 0.05 at the age of 58. These milestones are important in the planning of clinical trials that aim to prove that the disease has stopped after therapy. Ideally, patients would be treated just before the rapid decline of visual function, especially if the other eye was used as a control. For the p.Cys870* cohort, the optimal age to enter a clinical trial could be between 40 and 50 years of age. The other allele could also be of importance in patient selection for clinical trials. While all p.Cys870* patients shared a similar disease timeline, patients 3 and 4, who harbored a missense allele in trans with p.Cys970*, showed some indications of a milder disease. This was more notable for patient 4 harboring the variant p.Arg303His, a variant that has been previously noted to result in sparing of the peripheral retina in that and one other patient [10]. Preservation of the peripheral retina was also reflected in ffERG amplitudes, which were still detectable at the age of 52. It may take longer to show that the drug is working if eyes of patients with slower disease progression are used as controls. Another genetic aspect that needs to be considered is that homozygous patients for variants in exon 13 could respond best to therapy, as any effect would presumably be doubled.

### 3.2. Comparison of p.Cys870* Disease Progression with Other USH2A and Non-USH2A Retinitis Pigmentosa-Causing Genotypes

The rate of visual acuity loss of p.Cys870* patients (median 0.03 logMAR/year) was similar to that reported by Sandberg et al. on a large cohort of *USH2A* patients (0.026 logMAR/year) [12]. However, a severe drop in visual acuity occurred relatively early. The p.Cys870* patients’ visual acuity deteriorated to 0.1 or lower almost a decade earlier than was found in a multicentric syndromic *USH2A* study (55 vs. 64 years old) [13]. The difference could be due to the fact that most (78%) of the p.Cys870* patients in this study carried two null variants and were therefore presumably without a functional Usherin protein. In the multicentric study, the percentage of patients with double null variants was lower (51%, 105/204) [13] (Appendix A), therefore half of the patients could possibly have had some residual Usherin function. One of the two patients in the present study with a second missense variant indeed had significantly better-preserved visual acuity than the rest of the group, supporting this observation (Figure 1, Pt. 4). This genotype-phenotype correlation has also been found in a previous large study that compared 152 syndromic and 73 non-syndromic *USH2A* patients [14]. The syndromic patients more often harbored null variants and experienced severe visual loss more than a decade later than non-syndromic patients [14]. Accordingly, one of the most frequent *USH2A* variants, c.2276G > T (p.Cys759Phe), also residing in exon 13, has been shown to result in different phenotypes depending on the second allele, causing predominantly Usher syndrome type 2 when in *trans* with a null allele and predominantly isolated RP in homozygous state or in *trans* with another missense allele [15]. Although all patients in the present study had hearing loss, the p. Cys870* variant has also been found in patients with isolated RP, for example in a study by Comander et al. (2017) in a patient with pericentral RP who had two other missense variants [16]. Although in our cohort one patient also had pericentral RP, it is more likely that this phenotype is associated with the residual function of the missense variant.

Interestingly, in comparison with homozygotes for another null variant, p.Trp3955*, visual loss was earlier in p.Cys860* patients; 50% of them had a visual field < 20° at 52 years of age (vs. 59) and BCVA ≤ 0.05 at 58 years of age (vs. 66) [17]. This possibly suggests differences in the nonsense-mediated decay of different null variants, especially since p.Trp3955* is close to the 3’ end; however, larger cohorts and/or mRNA studies are needed to confirm this.

Comparing the disease progression with the second most frequent gene causing Usher syndrome, MYO7A [18], on average, MYO7A patients reach legal blindness at a younger age. In one study, 50% of MYO7A patients reached legal blindness based on either VF < 20° or BCVA ≤ 0.1 at 41 years of age [18], whereas in comparison, 50% of the p.Cys870* *USH2A* patients in the present study experienced the same visual loss at 50 years of age.

### 3.3. Advantages and Disadvantages of the Study

The main advantages of the study are the genetic homogeneity and the long follow-up. Retinitis pigmentosa is a slowly progressive disease, and changes are difficult to detect on a year-to-year basis. Most patients in our center have been followed-up on a regular basis for more than a decade, allowing us to determine the natural disease course. The disadvantage of the paper is that only the variant c.2610C>A (p.Cys870*) in exon 13 was included, which is a relatively rare exon 13 variant world-wide (30 reports in LOVD, available at databases.lovd.nl/shared/genes/USH2A, accessed on November 28, 2022). In other populations, c.2299delG (p.Glu767Serfs*21) (951 reports in LOVD) and c.2276G > T (p.Cys759Phe) (583 reports in LOVD) are much more frequent [9], but they are not present in the Slovenian cohort. Considering the above-mentioned differences, even between the presumably null genotypes, a precise knowledge of genotype-specific groups is of importance, and care is needed before assuming a similar disease course for other patients. One limitation of the study is the use of subjective methods to detect retinal function, i.e., visual acuity, Goldmann perimetry, and microperimetry. For example, up to 20% test-retest variability has been demonstrated for Goldmann perimetry, even when performed by the same technician [19,20,21]. Electrophysiology and pupillometry are functional tests that can provide more objective measures of retinal function [22]. They are unfortunately not being used routinely and longitudinally in clinical practice but are important in prospective clinical trials. Other reasons for the variability of the visual function can be changes in the optic media, such as cataract (e.g., in patient 8, Figure 1) and posterior capsule opacification; or CME (e.g., in patient 2, Figure 2). These factors need to be carefully considered in each patient before making conclusions regarding the trends of visual function deterioration, especially in a small patient cohort. None of the above-mentioned factors, however, affect the structural findings, such as the size of the hyperautofluorescent ring and the extent of the inner segment ellipsoid, which are therefore the most objective markers of retinal degeneration.

## 4. Materials and Methods

Nine Slovenian syndromic *USH2A* patients from seven families were included in our study (five females and four males). They all had a c.2610C>A (p.Cys870*) variant in exon 13 of the *USH2A* gene, two of which were homozygous and the other two were heterozygous. The majority of patients had been identified and genotyped in the scope of a multicentric study, Treatrush [11,23], and one relative was added subsequently (patient 9). Patient 4 has been included in a previous publication describing double hyperfluorescent rings in a subgroup of *USH2A* patients with preserved peripheral retina [10]. The median age at the first exam was 42 years (range, 25–56 years of age). Seven patients were followed longitudinally, with a median follow-up time of 11 years (range, 9–23 years of age). Disease onset was determined from medical history as the age when patients first noticed difficulties with night vision (nyctalopia). Visual acuity was measured with Snellen charts and converted to logMAR. Color vision was measured using Ishihara tables. Manual Goldmann perimetry (target II/4) was used to estimate the extent of the remaining central island of the visual field. The visual fields were scanned and measured using Image J (available at imagej.net). The remaining peripheral islands of the visual field (if present) were not included in this analysis to ensure better correspondence with the hyperautofluorescent ring measurement. For both, visual acuity and visual fields, Kaplan Meier survival analysis was performed to determine the age when 50% of patients reached legal blindness, based on the VA ≤ 0.1 (20/200) or VF diameter < 20° [12,18] in order to facilitate the comparison with other cohorts. Fundus autofluorescence imaging (FAF) and optical coherence tomography (OCT) were performed (Heidelberg, Spectralis) as previously described [11]. FAF patterns were then categorized into hyperautofluorescent rings, hyperautofluorescent patches and atrophy [11]. The accompanying software (Heidelberg, Spectralis) was used to measure the inner diameter of the hyperautofluorescenct ring. The presence of cystoid macular edema (CME) was determined based on OCT images. To analyze retinal function, static microperimetry was performed when possible, using Nidek MP1 (NAVIS software version 1.7.9, Nidek Technologies, Padova, Italy), after pupil dilatation with topical 1% tropiciamide. The sensitivity values from all 56 test loci and fixation were then superimposed over a 55° FAF image. Electrophysiology (ERG) data was reviewed if available. Five patients underwent full-field ERG, and three of them also underwent pattern ERG (PERG) with simultaneous visual evoked potentials (VEP), using the large field (30.7 × 23.6°) checkerboard. All recordings were performed according to the standards of the International Society of Clinical Electrophysiology of Vision (ISCEV) [24,25,26].

Yearly changes in visual acuity, visual field, and ring diameter, were calculated in patients with at least two measurements, by dividing the difference between the first and last exam with the number of follow-up years. The yearly loss was also calculated as a percentage of the initial value at the first exam. The disease was largely symmetrical between the eyes. The right eye was used for analysis except for Kaplan Meier, where the better-seeing eye was used to best represent the effect of disease on the patient.

The study was approved by the Slovenian Ethics Committee for Research in Medicine (55/01/11, 139/01/11), and all research procedures have been carried out in accordance with The Code of Ethics of the World Medical Association (Declaration of Helsinki) for experiments involving humans. Informed consent for genetic analysis was obtained from all subjects. The study was supported by the Slovenian Research Agency (ARRS J3-1750).

## 5. Conclusions

The study reports precise disease progression parameters relating to the *USH2A* variant p.Cys870*. The observations should be taken into account when planning clinical trials. We presume, that for p.Cys870* patients, successful treatment between the ages 40 and 50 would result in the most notable rescue of vision.

## Figures and Tables

**Figure 9 genes-14-00652-f009:**
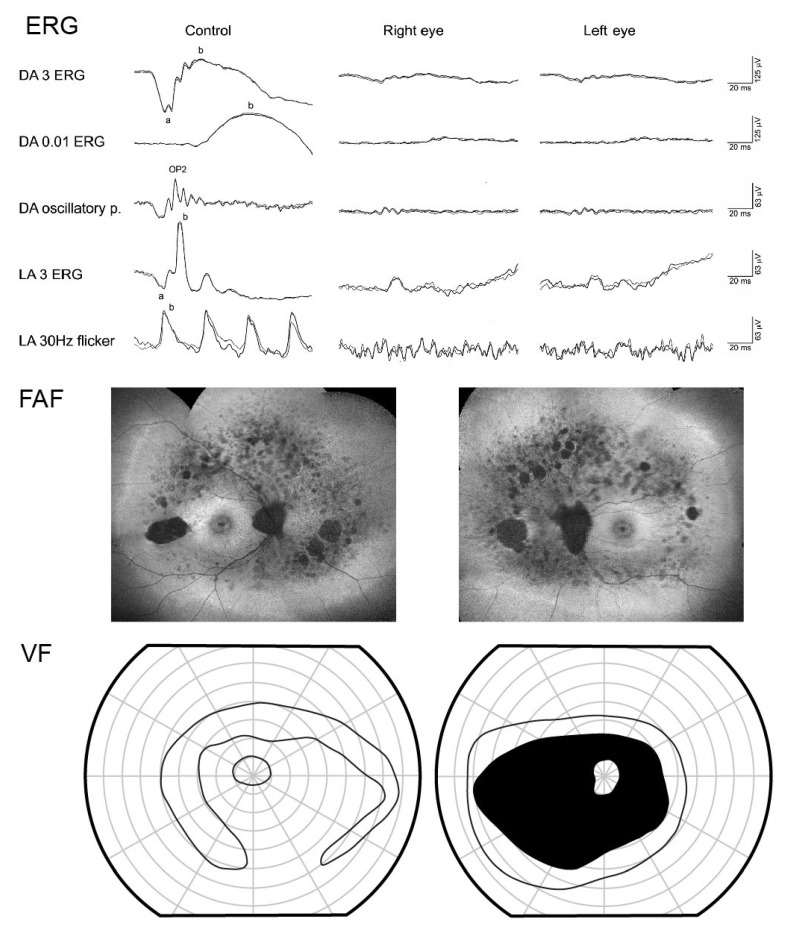
Electrophysiology (ERG), wide-field composite of fundus autofluorescence images (FAF), and Goldmann visual field (VF) of patient 4, harboring p.(Cys870*) and p.(Arg303His) at the age of 52. ERG showed absent DA 0.01 ERG (rod response) and reduced LA 3 ERG and LA 30 Hz flicker (cone response), typical for rod-cone dystrophy. There was some preservation of the peripheral retina noted on FAF and VF.

**Table 1 genes-14-00652-t001:** Genetic characteristics.

Patient	Center ID	Family ID	Allele 1	Allele 2	Genotype
1’	00526	0321	p.(Cys870*)	p.(Trp3955*)	Null
2’	00716	0321	p.(Cys870*)	p.(Trp3955*)	Null
3	00474	0381	p.(Cys870*)	p.(Thr4315Pro)	Non-null
4	00487	0389	p.(Cys870*)	p.(Arg303His)	Non-null
5	00903	0602	p.(Cys870*)	p.(Cys870*)	Null
6	00904	0603	p.(Cys870*)	p.(Trp3955*)	Null
7	00907	0606	p.(Cys870*)	p.(Cys870*)	Null
8”	00918	0617	p.(Cys870*)	p.(Trp3955*)	Null
9”	01050	0617	p.(Cys870*)	p.(Trp3955*)	Null

Sibling pairs are marked with ’ and ”, respectively.

**Table 2 genes-14-00652-t002:** Phenotypic characteristics of each patient at the first and last exam.

Patient	Sex	Age at Onset (Years)	Age at Exam (Years)	BCVA (Snellen Decimal)		Ishihara (N Out of 15 Plates Seen)		$\mathrm{Goldmann}\;\mathrm{II}4\;\mathrm{Area}\;(deg^{2});\;\mathrm{Diameter}\;(\deg)$		Microperimetry Mean Sensitivity (dB)		FAF Pattern (RE) (Age)	
				RE	LE	RE	LE	RE	LE	RE	LE	RE	LE
1’	F	35	42	0.5	0.6	9	9	238; 23	360; 20	2.4 (44)	2.9 (44)	Ring	Ring
			53	CF	CF	0	0	41; 7	109; 12	0.0 (54)	0.0 (54)	Patch	Patch
2’	F	8	43	N/A	N/A	N/A	N/A	343; 22	N/A	0.0 (54)	N/A	Patch (54)	N/A
			64	N/A	0.05	0	N/A	176; 15 (63)	N/A	N/A	N/A	Atrophy	N/A
3	F	21	30	0.25	0.16	N/A	N/A	8225; 114	7316; 108	1.6 (39)	3.6 (39)	Ring (43)	Ring (43)
			53	0.05	0.05	0	0	265; 18	665; 29 (48)	0.1	0.1	Patch	Patch
4	F	22	56	0.9	0.8	1	1	3838; 65 (57)	3477; 35 (57)	0.8	1.1	Ring	Ring
			65	CF	0.5	N/A	N/A	280; 20	194; 17	0.3	0.0	Ring	Ring
5	M	19	26	0.9	1	14	14	833; 40	N/A	2.6	2.6	Ring	Ring
			35	CF	0.8	N/A	N/A	N/A	N/A	0.6 (36)	1.4 (36)	Ring	Ring
6	F	11	49	CF	0.4	0	0	322; 18	390; 19	0 (57)	0 (57)	Atrophy	Patch
			64	HM	HM	N/A	N/A	256; 19 (60)	221; 18 (60)	N/A	N/A	Atrophy	Atrophy
7	M	23	25	1	1	13	13	426; 22	362. 22	2.9 (25)	N/A	Ring	Ring
8”	M	12	41	0.4	0.25	1	1	20; 5	29; 6	0.8	0.8	Ring	Ring
			52	0.3	0.4	N/A	N/A	N/A	N/A	N/A	N/A	Ring	Ring
9”	F	N/A	53	0.3	0.3	N/A	N/A	N/A	N/A	N/A	N/A	Ring	Ring

F = Female, M = Male, RE = Right Eye, LE = Left Eye, BCVA = Best corrected visual acuity; N/A = Not available, CF = Counting fingers. Pairs of siblings are marked with ’ and ”, respectively. Imaging and visual fields were not available for some patients at all exams; the age is noted in brackets if it differs from the stated age in column 4.

## Data Availability

All data is available in the Appendix A.

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
