# Peer review of "Natural Disease Course in Usher Syndrome Patients Harboring USH2A Variant p.Cys870* in Exon 13, Amenable to Exon Skipping Therapy"

_genes, 2023, doi:10.3390/genes14030652_

Round 1
Reviewer 1 Report
This is a good study of a small cohort of 9 Usher patients who all have mutations in the USH2A at p.Cis870*. This study is appropriately conducted. These are difficult studies because they must be retrospective over an extended period of time, and it is difficult to obtain Exactly identical measurements over time. The study definitely adds to the literature and is important contribution to rates of progress.
The following are recommendations for correction:
Line 19. Please define criteria for legal blindness by BCVA measurement.
Line 21. Add “radial.“ As in “… radial 1.5°/year…”
There are potentially many different ways to analyze these data, and I strongly recommend the authors make the full data set available for all measurements, so others can analyze in ways different from the study.
Author Response
We thank the reviewer for the valuable comments.
Line 19. Please define criteria for legal blindness by BCVA measurement.
Answer: We have defined the criteria as BCVA ≤0,1 Snellen decimal. We have corrected the wording for clarity and added the US/UK nomenclature (20/20). The sentence now states:
"Survival analysis predicted that 50 % patients reach legal blindness based on visual field diameter < 20° at the age of 52 years (95% CI, 45-59) and legal blindness based on BCVA ≤ 0,1 (20/20) at the age of 55 years (95% CI 46-66)."
We have the following statement to Methods: "For both, visual acuity and visual fields, Kaplan Meier survival analysis was performed to determine the age when 50% patients reached legal blindness, based on either VA ≤ 0,1 (20/20) or VF diameter < 20°, the definitions used in some of the previous studies [15,16] in order to facilitate the comparison with other cohorts."
Line 21. Add “radial.“ As in “… radial 1.5°/year…”
Answer: corrected as suggested
There are potentially many different ways to analyze these data, and I strongly recommend the authors make the full data set available for all measurements, so others can analyze in ways different from the study.
Answer: this is a valid argument. We agree it is difficult to choose which was of analysis is the most clear in presenting the data. We have prepared the datasets and included them in the supplement.
"We have added the following sentence at the beginning of the Results section: The summary of genetic and phenotypic characteristics is shown in Tables 1 and 2, respectively. The complete dataset of all measured clinical data is available in the Supplemental Table S1."
Reviewer 2 Report
Clinical course is evaluated to determine the rate of retinal degeneration in patients with p.Cys870*in USH2A exon 13, amenable to exon skipping therapy. Nine patients from seven families (3 male) were enployed (two homozygous). Seven patients had follow-up data (median of 11 years). The authors conclude that rates of progression will be useful in the design and execution of clinical trials.
Overall, the method of research is reasonable, and the research may be useful. However, there are concerns about the following points.
General points
1. In Case 2 at 54-year-old, OCT revealed macular edema. There was a period during which visual acuity seems to be improve slightly. As a limitation to the conclusions of this study, it is necessary to comment on complications in such individual cases.
2. It is useful to show the microperimetry of many cases over time. But, the small size of the figures makes it difficult to explain the authors' figure legend. I recommend to devise a way to decrease the figures or a explain it carefully.
3. Electroretinogram is not described. Some comments may be necessary.
4. I would like to include a little more of the previous literature in the discussion on the clinical picture of USH2A mutations including c.2276G > T.
Specific points
Page 9, line 195, “Ush2A” is “USH2A”.
Page 9, lines 222-225, although it is described as follows, I think it is necessary to describe how to handle the residual visual field left in the periphery.
Manual Goldmann perimetry (target II/4) was used to estimate the extent of peripheral visual field. The visual fields were scanned and measured using Image J (available at imagej.net).
Author Response
We thank the reviewer for the valuable comments.
- In Case 2 at 54-year-old, OCT revealed macular edema. There was a period during which visual acuity seems to be improve slightly. As a limitation to the conclusions of this study, it is necessary to comment on complications in such individual cases.
Answer: Thank you for the valuable suggestion. We believe the comment is referring to the improvement of visual field and not visual acuity (Fig. 2) (Please correct us if otherwise).
We have added the following comment under the Fig. 2.: "There is an apparent improvement of visual field of Patient 2 after previous worsening. The patient had extensive cystoid macular edema CME around the age of 50 years (Fig. 5) which resolved after time and may have contributed to the variability of the visual field. "
We have also added a paragraph on the variability of the subjective measurements of the visual function in Discussion: "One limitation of the study is the use of subjective methods to detect retinal function, i.e., visual acuity, Goldmann perimetry and microperimetry. For example, up to 20 % test-retest variability has been demonstrated for Goldmann perimetry, even when performed by the same technician [17-19]. Electrophysiology and pupillometry are functional tests that can provide more objective measures of retinal function [20]. They are unfortunately not being used routinely and longitudinally in clinical practice but are important in prospective clinical trials. Other reasons for the variability of the visual function can be changes in the optic media, such as cataract (e.g., in Patient 8, Fig. 1) and posterior capsule opacification; or CME (e.g., in Patient 2, Fig. 2). These factors need to be carefully considered in each patient before making conclusions regarding the trends of visual function deterioration, especially in a small patient cohort. None of the above-mentioned factors however affect the structural findings, such as the size of the hyperautofluorescent ring and the extent of the Inner segment ellipsoid, which are therefore the most objective markers of retinal degeneration.
- It is useful to show the microperimetry of many cases over time. But, the small size of the figures makes it difficult to explain the authors' figure legend. I recommend to devise a way to decrease the figures or a explain it carefully.
Answer: Thank you for the comment. Since the cohort is small, we think it is valuable to include all available imaging and microperimetry data. However, we agree the microperimetry + FAF images are probably too small. Therefore, we have added two more Figures with enlarged microperimetry and arranged patients in a way to provide a better depiction of the progression of disease with age as well as demonstrate genotypic differences (milder phenotype in one patient with a missense variant). An attenuation scale of MP is added to these figures for clarity.
- Electroretinogram is not described. Some comments may be necessary.
Answer: Thank you for the comment. We have included a paragraph on ERG in the relevant sections of Methods, Results and Discussion.
Methods: " Electrophysiology (ERG) data was reviewed if available. Five patients underwent full-field ERG and three of them also Pattern ERG (PERG) with simultaneous visual evoked potentials (VEP), using the large field (30.7 x 23.6°) checkerboard). All recordings were performed according to the standards of the International Society of Clinical Electrophysiology of Vision (ISCEV) [22-24]."
Results: "2.5. Electrophysiology. Five patients underwent full-field electrophysiology (ffERG) testing. The ffERG amplitudes in all cases were consistent with retinitis pigmentosa. They were reduced in the pattern of rod-cone dystrophy in two patients (P4 and P5; ages 52 and 26 years, respectively). and undetectable in three patients (P1, P6 and P7; ages 42, 50, and 25 years, respectively). The oldest patient with detectable ffERG was Patient 4, at the age of 52, who harbored p.(Arg303His) on the second allele. Patients P1, P4 and P5 also underwent pattern ERG (PERG) and visual evoked potentials (VEP) at ages 42, 52 and 26 years, respectively. All had reduced PERG P50 amplitude and normal VEP amplitude and latency. Their visual acuity at the time of the recording was 0.5 - 1.0 Snellen decimal and all three exhibited hyperautofluorescent ring on FAF at that time (or at the first occasion that imaging was performed, if performed subsequently)."
Discussion: "While all p.Cys870* patients shared a similar disease timeline, the patients 3 and 4, who harbored a missense allele in trans with p.Cys970*, showed some indications of a milder disease. This was more notable for the Patient 4 harboring the variant p.Arg303His, a variant that has been previously noted to result in sparing of the peripheral retina in that and one other patient [11]. Preservation of the peripheral retina was also reflected in ffERG amplitudes, that were still detectable at the age of 52 years."
- I would like to include a little more of the previous literature in the discussion on the clinical picture of USH2A mutations including c.2276G > T.
Answer: Thank you for the suggestion. We have expanded the discusion with and added information on the clinical picture of USH2A mutations, including c.2276G > T.
Specific points
Page 9, line 195, “Ush2A” is “USH2A”. Corrected
Page 9, lines 222-225, although it is described as follows, I think it is necessary to describe how to handle the residual visual field left in the periphery.
Manual Goldmann perimetry (target II/4) was used to estimate the extent of peripheral visual field. The visual fields were scanned and measured using Image J (available at imagej.net).
Answer: The Methods section was amended to clarify how the measurement of the visual field was performed: "Manual Goldmann perimetry (target II/4) was used to estimate the extent of the remaining central island of the visual field. The visual fields were scanned and measured using Image J (available at imagej.net). The remaining peripheral islands of visual field (if present) were not included in this analysis to ensure better correspondence with the hyperautofluorescent ring measurement."
Round 2
Reviewer 2 Report
Figure 7, the font of "Ring...Patch...Atrophy" on the far left is considered inappropriate for an academic journal.
Author Response
Figure 7, the font of "Ring...Patch...Atrophy" on the far left is considered inappropriate for an academic journal.
Answer: Thank you for the comment. We have changed the font to Arial.
Yours sincerely
Ana Fakin
